# *Vibrio cholerae* serotype impacts pathogenicity

Franz G. Zingl [1,2,3], Deborah R. Leitner[1,2,3], Bolutife Fakoya[1,2,3], Alexander A. Morano[1,2,3] & Matthew K. Waldor [1,2,3] ✉

The O1 serogroup of *Vibrio cholerae* has caused all cholera pandemics and for over a century *V. cholerae* O1 outbreak strains have been characterized by their serotype. The two *V. cholerae* serotypes differ by the presence (Ogawa) or absence (Inaba) of methylation of the terminal sugar on the lipopolysaccharide O1-antigen. Serotype switching often occurs during epidemics and has historically been attributed to the pathogen adapting to immune pressures. Here we address the impact of serotype on *V. cholerae* pathogenicity using otherwise isogenic Ogawa and Inaba versions of several clinical *V. cholerae* O1 isolates. Our findings indicate that O1 antigen methylation in Ogawa strains promotes *V. cholerae* colonization, infectivity and resistance to antimicrobial peptides. We propose that methylation of the O1 antigen elevates colonization by shielding the bacterium from cationic antimicrobial peptides at the pH of the small intestine. These observations provide insights into the biological significance of the *V. cholerae* O1 serotypes.

Seven pandemics of cholera have been recorded over the past two centuries[1]. In recent years, cholera cases and fatalities have been increasing in many parts of the world due to several factors, including climate change and political and social conflicts that have led to the degradation of public health measures that provide sanitation and clean water[2]. All seven cholera pandemics are thought to have been caused by the O1 serogroup of *Vibrio cholerae*[3,4]. Serogroups are determined by the chemical composition of the O-antigen of lipopolysaccharide (LPS), the most distal part of LPS extending out from the cell surface. In *V. cholerae* O1 the O-antigen is a polymer of perosamine (Fig. 1a).

Two major serotypes - Ogawa and Inaba - of clinical isolates of the O1 serogroup have been recognized for more than a century. The serotypes can be distinguished based on their differential agglutination by specific antisera. Classification of clinical *V. cholerae* isolates as Ogawa or Inaba using such antisera has been routine for decades and remains common practice in clinical microbiology labs. The chemical basis for the difference in the serotypes has been elucidated and resides in the presence (Ogawa) or absence (Inaba) of methylation of the terminal perosamine in the O1 O-antigen (Fig. 1a).

The *wbeT* methyltransferase methylates the terminal perosamine, and Inaba strains contain a variety of inactivating mutations in *wbeT*[5]. Many sero-epidemiologic studies have shown that *V. cholerae* undergoes serotype switching during cholera epidemics[6–9], with the transition from the Ogawa to the Inaba serotype during epidemics being more common.

The O-antigen is the major protective antigen against cholera in humans[10–12]. Although there is often cross-reactivity in antibody binding between the serotypes, serotype switching during epidemics is generally thought to arise due to selective pressures exerted on the pathogen by the adaptive immune system. However, the impact of serotype on pathogenicity is largely uncharted.

Here, we generate pairs of otherwise isogenic Inaba and Ogawa strains and compare their ability to colonize the small intestine and resist host-derived antimicrobial factors. We show that Ogawa strains have a greater capacity for colonization and greater infectivity than Inaba strains. Mechanistically, these properties may be attributable to the greater resistance of Ogawa strains to cationic antimicrobial peptides. These observations introduce a new perspective on the significance of the *V. cholerae* serotypes.

[1]Division of Infectious Diseases, Brigham & Women's Hospital, Boston, MA, USA. [2]Department of Microbiology, Harvard Medical School, Boston, MA, USA. [3]Howard Hughes Medical Institute, Boston, MA, USA. ✉e-mail: mwaldor@research.bwh.harvard.edu

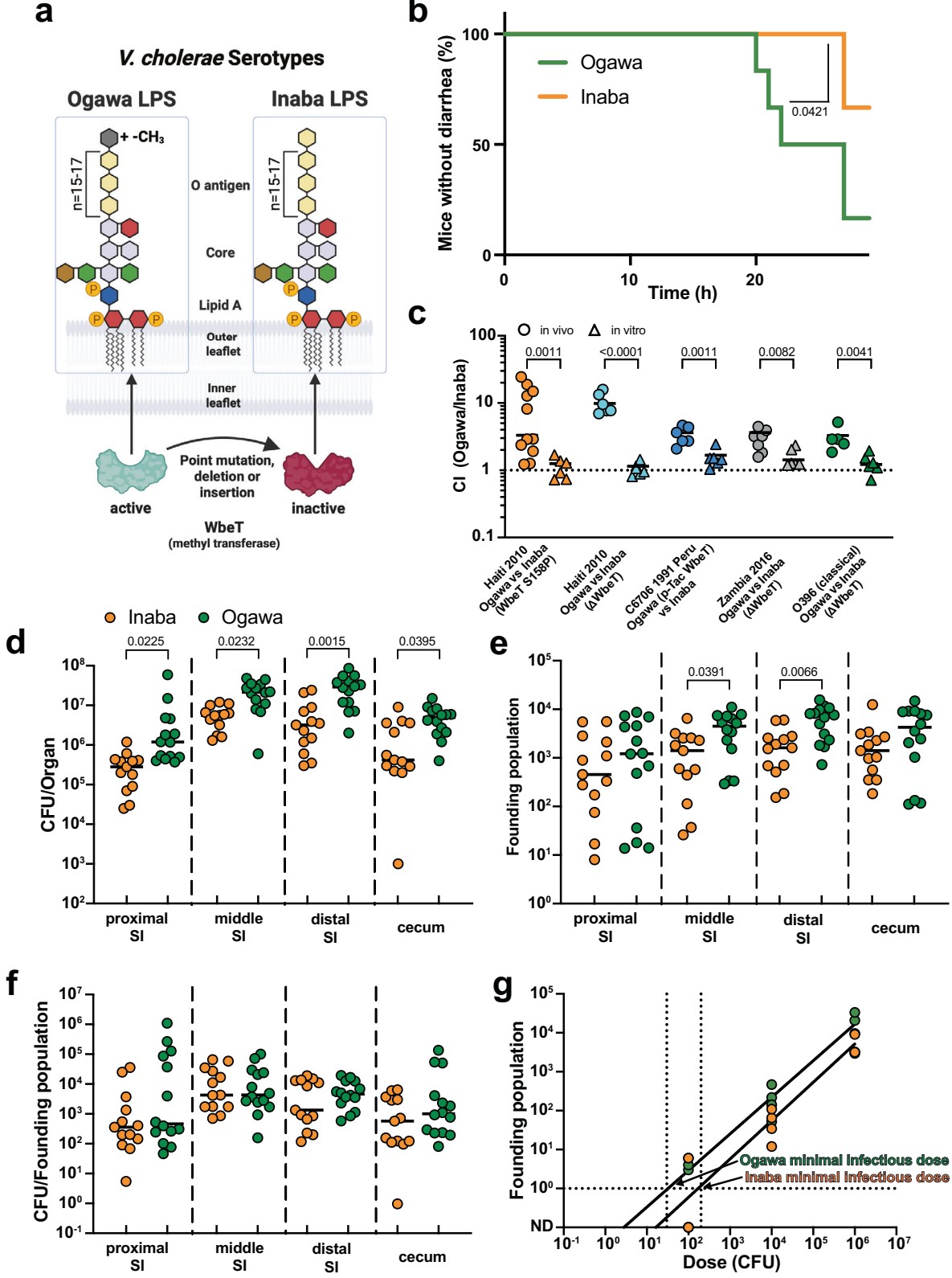

## Results

### Serotype impacts virulence and colonization

To test if isogenic Ogawa and Inaba strains cause similar diarrheal disease, we created an Inaba derivative of the Ogawa *V. cholerae* O1 strain that caused the 2010 cholera outbreak in Haiti. The only genomic difference between the strains is an S158P inactivating point mutation in *wbeT* (Fig. 1a). Unexpectedly, in infant mice, the Ogawa

strain caused diarrhea earlier and in more mice than the Inaba strain (Fig. 1b, Supplementary Fig. 1). This was accompanied by a trend toward more weight loss and greater diarrheal discharge (Supplementary Fig. 2). However, we did not observe differences in cholera toxin production in virulence activating conditions (AKI) or higher expression of either of the *ctxAB* subunits during in vivo colonization in the isogenic Inaba and Ogawa strains (Supplementary Fig. 3). These

**Fig. 1 | *V. cholerae* serotype influences virulence, colonization and infectivity.** **a** *V. cholerae* O1 is divided into Ogawa and Inaba serotypes. The terminal perosamine of the O-antigen of the LPS is methylated in the Ogawa serotype (grey hexagon) and unmethylated in the Inaba serotype (yellow hexagon). **b** Percentage of suckling mice that have no diarrhea in the first 24 h after infection with Inaba (orange) or Ogawa (green) *V. cholerae* serotype ($n = 6$). Significance calculated with two-sided Mantel-Cox test. **c** Competitive indices (CI) of Inaba and Ogawa strains in suckling mice (circles) and LB (triangles) after 18 h. CI calculated as ratio of Ogawa to Inaba CFU divided by the ratio in the inoculum for the following pairs: Haiti 2010 Ogawa vs Inaba WbeT S158P (orange), Haiti 2010 Ogawa vs Inaba Δ*wbeT* (light blue), C6706 1991 Peru Inaba vs Ogawa p-Tac WbeT (blue), Zambia 2016 Ogawa vs Inaba Δ*wbeT* (grey) and O395 (classical) Ogawa vs Inaba Δ*wbeT* (green) ($n = 6$, except for Haiti 2010 Ogawa vs Inaba (Wbet S158P) in vivo $n = 11$, Zambia 2016 Ogawa vs Inaba Δ*wbeT* in vivo $n = 7$ and O395 (classical) Ogawa vs Inaba (Δ*wbeT*) in vivo $n = 5$). Significance calculated with two-sided unpaired T test with Welch's correction of log transformed data. **d** CFU of isogenic Inaba (orange) or Ogawa

(green) strains in the indicated region of the intestine ($n = 13$ for Inaba and $n = 14$ for Ogawa). Significance calculated with two-sided Kruskal-Wallis test followed by Dunns uncorrected multiple comparisons. **e** Founding population sizes of isogenic Inaba (orange) or Ogawa (green) strains in the indicated regions of the intestine ($n = 13$ for Inaba and $n = 14$ for Ogawa). Significance calculated with a two-sided Kruskal-Wallis test followed by Dunns uncorrected multiple comparisons. **f** Intestinal replication of isogenic Inaba (orange) or Ogawa (green) strains, calculated by dividing CFU by founding population ($n = 13$ for Inaba and $n = 14$ for Ogawa). **g** Infectivity of isogenic Inaba (orange) or Ogawa (green) strains, calculated by estimation of minimal infectious dose, which is the dose at which 1 bacterium would pass the bottleneck and establish infection ($n = 3$ for Inaba $10^2$ and $10^6$; otherwise, $n = 4$ per dose). All mice used were Crl:CD1(ICR) mixed sex, postnatal day 5 at the time of infection. Unless specified, Ogawa denotes a 2010 Haiti isolate[30], and Inaba the isogenic *wbeT* S158P mutant. In all cases, n represents biological replicates. Source data are provided as a Source Data file. Parts of this figure were created in BioRender. Zingl, F. (2025) https://BioRender.com/u7tr1gr.

findings suggest that the observed greater Ogawa virulence may result from higher colonization.

We used the infant mouse model to compare the capacity of isogenic Ogawa and Inaba strains to colonize the small intestine. When ~1:1 mixtures of the Ogawa and Inaba strains were inoculated into mice, greater Ogawa vs Inaba colony forming units (CFU) were recovered from intestinal homogenates. In contrast, when the mouse inoculum was expanded in LB media, the Ogawa and Inaba strains grew equivalently, and the ratio of the strains remained ~1:1 (Fig. 1c). Very similar results were obtained with 4 other strain pairs: another Inaba derivative of the Ogawa Haiti clinical isolate entirely lacking *wbeT* (Δ*wbeT*), a complementation of *wbeT* in a 1991 Inaba clinical isolate from Peru (C6706), a Δ*wbeT* Inaba derivative of a 2016 Zambian Ogawa clinical isolate, and an Inaba version (Δ*wbeT* mutant) of the 1965 classical biotype *V. cholerae* O1 Ogawa isolate O395 (Fig. 1c, Supplementary Fig. 4). These observations reveal that in both *V. cholerae* biotypes, Ogawa strains have a greater capacity for intestinal colonization than isogenic Inaba strains, suggesting that the methylation of the terminal perosamine in the O1 O-antigen enhances *V. cholerae's* capacity to colonize the intestine.

### Ogawa serotype is more infectious than Inaba

The infection-related population dynamics that resulted in the superior colonization of the Ogawa strains were investigated using barcoded libraries of otherwise isogenic Ogawa and Inaba strains. The barcodes enable calculation of founding population sizes, the number of cells from the inoculum that give rise to the CFU observed in a sample[13] (Supplementary Fig. 5). Infant mice were inoculated with either the barcoded Ogawa or Inaba libraries. Greater numbers of Ogawa CFU were recovered and the size of the Ogawa founding population was greater in all samples (Fig. 1d,e). In contrast, there was no difference between the serotypes in CFU/Ns (Fig. 1f), a measure of net replication per founder, suggesting that the number of cells originating from one founder is similar between the two serotypes. Collectively, these data suggest that the greater colonization of the Ogawa serotype is largely attributable to its greater capacity to establish a replicative niche (higher founding population) in the intestine compared to the Inaba serotype; once such niches are established, both serotypes appear to have similar capacities for in vivo replication.

Differences in founding population sizes can reflect differences in pathogen infectivity, which are often measured as the dose at which 50% of hosts get infected ($ID_{50}$). To evaluate if this is the case for the two serotypes, we infected suckling mice with barcoded libraries at different doses. The X-intercept of the line depicted the relationship between the founding population vs the inoculum size provides an estimate of the minimal infectious dose that corresponds to the $ID_{50}$[13]. The minimal infectious dose was approximately 10 times lower for

Ogawa than Inaba strains, demonstrating their elevated infectivity, i.e., their greater inherent ability to infect hosts (Fig. 1g).

### Serotype impacts resistance to cationic antimicrobial peptides
To identify the mechanistic bases for the different founding population sizes, we used a set of barcoded Ogawa and Inaba strains to simultaneously compare the capacity of the two serotypes to survive and grow in several different potential in vivo stressors. We first corroborated the finding that Ogawa strains have a greater capacity for intestinal colonization using this system (Fig. 2a). Then, different stressors that may be confronted in the intestine, including pH 6, pH 8, pH 8.5, bile, polymyxin B (an antimicrobial peptide), intestinal fluid, and human serum, were tested. Relative growth of the two serotypes in a variety of laboratory media and in most of the stressors tested was similar (Fig. 2b). However, the Ogawa serotype outcompeted Inaba in the presence of polymyxin B (Fig. 2b).

Polymyxin B is a cationic antimicrobial peptide (AMP) that binds to the LPS of Gram-negative bacteria due to its charge[14]. Since the charge of molecules is affected by pH, we speculated that its interactions with and killing of bacteria may differ depending on pH. We compared whether the sensitivity of isogenic Ogawa and Inaba strains to polymyxin B changes with pH. In the absence of polymyxin B, the growth of the serotypes was indistinguishable (Fig. 2c, d). However, in polymyxin B, the growth of the Inaba strain was more impaired than the Ogawa strain at pH 8 (Fig. 2c), but not at pH 6 or 7 (Supplementary Fig. 6). Furthermore, we found there was no difference in growth of the Ogawa and Inaba strains at each of the 3 pHs tested in the absence of AMPs. However, at pH 8 but not at pH 6 or 7, Ogawa strains grew better than Inaba strains in the presence of polymyxin B and the human intestinal AMP LL-37 (Fig. 2d), indicating that methylation of the terminal perosamine in Ogawa LPS provides resistance to cationic AMPs at pH 8. Consistent with the hypothesis that AMP resistance is pH dependent, we found that when the pH of the in vivo inoculum was buffered to pH 6 or pH 7, Ogawa lost its competitive advantage over Inaba (Fig. 2A).

To test if methylation alters binding of AMPs to the outer membrane in a pH dependent fashion, isogenic Ogawa and Inaba cells were treated with fluorescently labeled polymyxin B and LL-37. At pH 8, binding of the labeled polymyxin to Inaba cells was readily observed but hard to detect on Ogawa cells, whereas at pH 7 no difference in the binding was detectable (Fig. 2e). Measurements of fluorescence intensity revealed no difference between Inaba and Ogawa at pH 7, whereas at pH 8 fluorescence was significantly greater on Inaba vs Ogawa cells for both AMPs (Fig. 2f). These findings suggest that methylation of the O-antigen in Ogawa cells can block or reduce the interaction between cationic antimicrobial peptides and LPS.

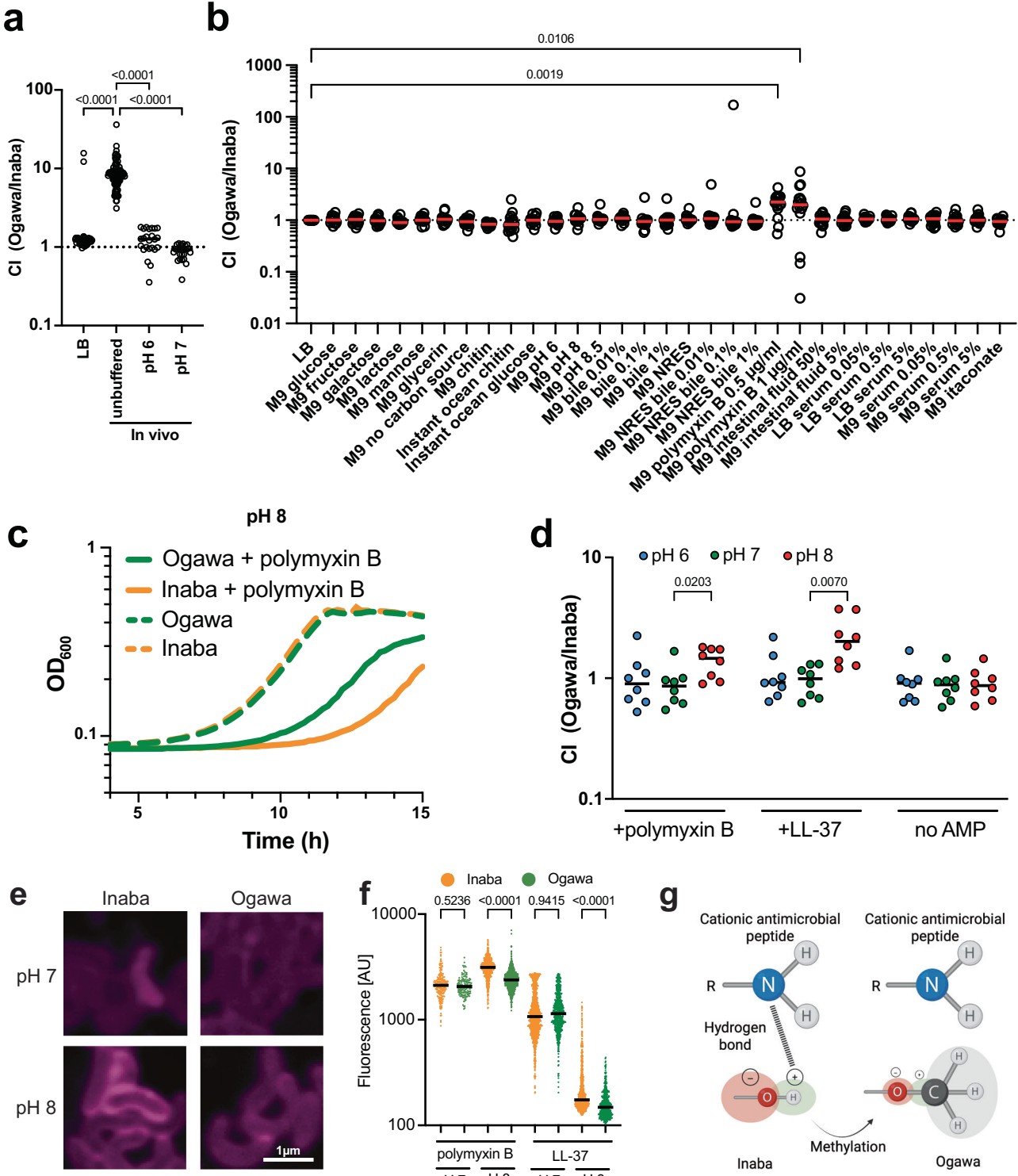

## Discussion

Our findings revealed that *V. cholerae* serotype impacts pathogenicity. We hypothesize that by altering the pathogen's resistance to cationic antimicrobial peptides at pH 8, the Ogawa serotype attains a higher capacity to colonize the intestine. Accordingly, we propose that the elevated virulence observed during Ogawa infection results from its enhanced colonization capacity. To explain the alkaline pH-dependent sensitivity of Inaba strains to cationic antimicrobial peptides, we favor a model where at pH 8, the ammonium ($NH_3^+$) groups on cationic antimicrobial peptides are largely present as amino ($NH_2$) groups. These amino groups can form hydrogen bonds

with the hydroxyl (OH) of the Inaba terminal perosamine but not to the methoxy ($OCH_3$) group of the terminal perosamine in the Ogawa serotype (Fig. 2g).

Our observations reveal that the *V. cholerae* Ogawa serotype has a greater capacity for intestinal colonization and virulence than the Inaba serotype. Notably, Ogawa strains are also thought to be more virulent in humans[15], however studies with isogenic Ogawa/Inaba strain pairs have not been carried out in humans. Review of the literature suggests that most[16–18] though not all[19] cholera outbreaks begin as the Ogawa serotype. Our findings that Ogawa has a lower infectious dose, higher colonization capacity, and likely greater pathogen load in the

**Fig. 2 | Multiplex screening of the growth of barcoded isogenic Ogawa and Inaba strains in vivo and in a variety of growth conditions in culture.** Ogawa and Inaba strains with known unique genetic barcodes were mixed in a 1:1 ratio and then either **a** used to infect suckling mice with or without a buffered inoculum ($n = 46$ for LB, $n = 92$ for in vivo and $n = 24$ for each buffered in vivo experiment; Significance calculated with two-sided Kruskal-Wallis test followed by Dunn's correction for multiple comparisons) or **b** grown in presence of specific nutrients or stressors. All media types are unbuffered if not stated otherwise. The ratio of the abundance of Ogawa and Inaba barcodes after 20 hours is used to calculate a competitive index (CI). Each point represents a comparison between Ogawa and Inaba ($n = 16$). Significance calculated with two-sided Kruskal-Wallis test followed by Dunn's correction for multiple comparisons. **c** Growth curves at pH 8 of Inaba (orange) or Ogawa (green) strains in M9 with (solid line) or without (dashed line) polymyxin B ($n = 3$). **d** Barcoded libraries were treated with cationic antimicrobial peptides polymyxin B or LL-37 at pH 6 (blue), pH 7 (green), and pH 8 (red) and the relative abundance of Ogawa and Inaba serotypes after 20 hours is shown ($n = 8$; $n$ represent competitions between independent barcodes of the Ogawa and the Inaba serotype). Significance calculated with a two-sided Kruskal-Wallis test followed

by an unadjusted Dunn's post hoc test. **e** Representative images of Inaba and Ogawa strains treated with rhodamine labeled polymyxin B at pH 7 and pH 8. **f** Relative intensity of rhodamine B labeled polymyxin B at pH 7 ($n = 206$ for Inaba and $n = 148$ for Ogawa) and pH 8 ($n = 721$ for Inaba and $n = 1217$ for Ogawa) as well as rhodamine B labeled LL-37 bound to Inaba (orange) or Ogawa (green) cells at pH 7 ($n = 646$ for Inaba and $n = 1134$ for Ogawa) and pH 8 ($n = 943$ for Inaba and $n = 499$ for Ogawa). Significance calculated with a two-sided Kruskal-Wallis test followed by an unadjusted Dunn's post hoc test. **g** Proposed model for differential binding of polymyxin B to Ogawa vs Inaba LPS. Methylation of the O-antigen leads to a blockage of possible hydrogen bonds at pH 8 between the Inaba terminal perosamine OH group and the $NH_2$ groups of the cationic antimicrobial peptide. All mice used were Crl:CD1(ICR) mixed sex, postnatal day 5 at the time of infection. Unless specified, Ogawa denotes the 2010 Haiti isolate[30], and Inaba the isogenic *wbeT* S158P mutant. Significance calculated with two-sided Kruskal-Wallis test followed by Dunn's correction (**a**, **b**) or uncorrected (**d**, **f**) for multiple comparisons. In all cases $n$ represents biological replicates. Source data are provided as a Source Data file. Parts of this figure were created in BioRender. Zingl, F. (2025) https://BioRender.com/kits1dh

---

diarrheal discharge could explain this bias towards Ogawa strains initiating outbreaks.

Our observations also expose an intriguing evolutionary paradox: why do Inaba strains persist if they are less infectious and less efficient at intestinal colonization than Ogawa strains? It is possible that Inaba strains are fitter than Ogawa strains in some environmental niche, like growth on chitin or in confronting potential environmental predators such as ameba or phages[20,21]. Alternatively, the pressures of the human adaptive immune responses targeting Ogawa vs Inaba LPS may differ. It has been reported that Ogawa infections lead to less cross-protective immunity than Inaba infections[22]. It is possible that during periods of Ogawa dominance, Inaba-specific immunity is lost, thus rendering the population susceptible to this less infectious serotype.

## Methods
### Mice
All animal studies were conducted in compliance with the Guide for the Care and Use of Laboratory Animals at the Brigham and Women's Hospital and according to protocols reviewed and approved by the Brigham and Women's Hospital Institutional Animal Care and Use Committee (protocol 2016N000416). CDI dams with litters (mixed sex) were purchased from Charles River Laboratories (strain #022). All mice were housed in a biosafety level 2 (BSL2) facility under specific pathogen-free conditions at 68-75 °F, with 30-50% humidity, and a 12 h light/dark cycle. Infant mice were euthanized by isoflurane inhalation followed by decapitation. Adult mice were euthanized at the end of the study by isoflurane inhalation followed by cervical dislocation.

### Biosafety statement
All work on *V. cholerae* was performed in Biosafety Level 2 (BSL2) facilities at the Brigham and Women's Hospital according to protocols reviewed and approved by the Brigham and Women's Hospital Institutional Biosafety Committee (protocol 2011B000082). All personnel working with bacteria were trained in relevant safety and protocol-specific procedures.

### Bacterial strains and growth conditions
Bacteria were generally grown at 37 °C in lysogeny broth (LB) in liquid culture shaking at 200 rpm, on solid media containing 1.5% agar (weight/vol.), or in minimal medium M9 with 0.2% of the carbon source. If not stated otherwise, glucose was used as a carbon source. For cholera toxin induction, *V. cholerae* was grown in AKI broth (0.5% sodium chloride, 0.3% sodium bicarbonate, 0.4% yeast extract, and 1.5% Bacto peptone) for 4-hours anaerobically, followed by 4-hours

with shaking at 200 rpm at 37 °C with aeration[23,24]. When appropriate, antibiotics or other supplements were used in the following concentrations: streptomycin (Sm), 200 μg/ml; carbenicillin (Cb), 100 μg/mL; kanamycin (Km), 50 μg/mL; diaminopimelic acid (DAP), 0.3 mM; sucrose (Suc), 0.2%; 5-bromo-4-chloro-3-indolyl-β-d-galactopyranoside (X-gal), 60 μg/mL. Bacterial stocks were stored at -80 °C in LB containing 25-35% (vol./vol.) glycerol.

### Infant mouse diarrhea and colonization assays
Crl:CD1(ICR) infant mice at postnatal day 5 were inoculated intragastrically with the indicated dose and strain of *V. cholerae* in 50 μL of LB containing green food dye. The infected infant mice were housed separately from the dams in tissue-lined boxes. For colonization assays, infant mice were separated from their dams and intragastrically inoculated with the indicated strain at ~$2 \times 10^5$ cells or the indicated dose. If the inoculum was buffered, cells were diluted in 1 M Tris for pH 7 and 1 M MES for pH 6. Dose (colony forming units, CFU) was determined by serial dilution and plating of the inocula on selective media containing Sm. Sixteen to twenty hours post inoculation, pups were euthanized, and the SI was removed. Colonization burden was determined in either the whole SI, or the SI divided into 3 parts of equal length (proximal, medial, and distal) and the cecum. Intestinal segments were homogenized in LB using 2 stainless-steel 3.2 mm beads with a bead beater (BioSpec Product, Inc) for 2 minutes at 2700 rpm. CFU in intestinal homogenates was determined by serial dilution and plating on selective media containing Sm. To measure diarrhea, 5-day-old infant mice were inoculated with $2 \times 10^5$ bacteria and individually housed for 24-hours. Bedding material for each mouse was observed for stains of diarrheal discharge (green food dye). Competitive infections were performed in a similar manner, except that the inoculum was a ~1:1 mixture of the indicated strains or contained 8 barcoded Ogawa and 8 barcoded Inaba strains. The mixture was also used to inoculate LB cultures at a 1:1000 dilution. Blue/white CFU were determined by plating organ homogenates or cultures on LB + Sm/X-gal. Barcoded strains were plated on LB + Km and incubated for 16 h at 37 °C before being used for sequencing. Competitive indices were calculated by dividing the blue: white ratio in intestinal homogenates by the blue: white ratio from the inoculum or by dividing the Ogawa: Inaba barcode ratio in the homogenates by the Ogawa: Inaba barcode ratio of the inoculum.

### Growth kinetics
Growth kinetics were performed in transparent 96-well plates (Greiner) with 150 μl culture volume. The respective strains were grown in a pre-culture for ~16 h in LB with aeration and shaking at 37 °C. For growth assays, pre-cultures were diluted 1:10,000 in M9 at pH 7 or pH 8 with or without 2.5 μg/ml polymyxin B. The $OD_{600}$ was

monitored every 10 min at 37 °C with shaking. For presentation of data, the mean of at least three independent growth curves were plotted.

## Cholera toxin ELISA

A GM1 ELISA was used to quantify the concentration of CT in cell-free supernatant samples[25]. Ninety-six-well polystyrene microtiter plates were coated with GM1 ganglioside overnight, and 1% (wt/vol) fatty acid-free bovine serum albumin (BSA) was used to block the GM1-coated plates for 1 h at room temperature. Next, 260 μl of culture supernatant was added to the wells and incubated for 1 h at room temperature. Subsequently, a rabbit anti-CT polyclonal antibody (1:10,000, Abcam Cat. Nr.: 123129, lot: GR3254532-2) was added and incubated at 1 h at room temperature; then, an HRP-linked goat ani-rabbit IgG antibody (1:2,000, Sigma Cat. Nr.: A4914, lot: 061M6281) was added to the wells and incubated for 1 h at room temperature. For the development of the CT-antibody complex, tetramethylbenzidine (TMB) substrate solution (Thermo Fisher Scientific) was used according to the manufacturer's protocol. The color intensity in each well was measured at 485 nm in a plate reader. CT amounts in the samples were estimated by comparison to the standard curve constructed with known concentrations of purified CT (Sigma C8052).

## Cholera toxin qRT-PCR

Expression of *ctxA*, and *ctxB* was determined by quantitative real-time RT-PCR (qRT-PCR). Intestinal samples from mice inoculated with Ogawa or Inaba strains were obtained 16 hours after infection and homogenized in TRIZOL reagent. RNA was extracted using chloroform extraction and precipitated with isopropanol and subsequently used for PCR with Luna universal one-step RT-qPCR kit (NEB). Relative gene expression comparisons were obtained through the ΔΔCT method by normalizing the mean cycle threshold of each investigated transcript to the 16 s RNA and then normalized to a randomly selected sample.

## Construction of in-frame deletion mutants, Inaba point mutation, and Ogawa complementation strains

Constructions of in-frame deletion mutants of *wbeT* and *lacZ* as well as *wbeT* point mutation were carried using standard allele exchange with pCVD442 containing ~800 bp PCR fragments upstream and downstream of the target region[26–28] (Table S1). Briefly, respective pCVD442 were transformed into *E. coli* SM10λpir and conjugated into *V. cholerae*. Exconjugants were purified by selection as SmR/ApR colonies. Sucrose selection was used to obtain ApS colonies, and chromosomal deletions/replacements were confirmed by PCR and sequencing. Point mutation was confirmed via Sanger sequencing. To render the *V. cholerae* Inaba strain C6707 Ogawa, a derivative of the suicide vector pCVD442 containing *wbeT* under control of the constitutive pTac promoter was created and transformed into the SM10λpir *E. coli* strain. To create the pCVD442::pTac wbeT plasmid, a gene block containing the promotor, the wbeT gene, and the ~800 bp fragments upstream and downstream of the target region as well as the respective cut sides, was ordered (IDT) and cloned into a cut pCVD442 vector. After conjugation with *V. cholerae* strain C6706, integration into the STAMP locus was confirmed by PCR and slide agglutination.

## Construction of barcoded *V. cholerae* libraries

Barcoded libraries of Inaba and Ogawa *V. cholerae* were constructed as described[27]. Briefly, Inaba and Ogawa *V. cholerae* were conjugated with the *E. coli* donor strains MFDλpir pSM1 (barcode donor) and MFDλpir pJMP1039 (containing the helper Tn7 transposase) (Table S1). Transconjugants were selected on LB + Sm + Km plates, pooled in LB glycerol, and stored at −80 °C in aliquots.

## In vitro barcoded competition assays

8 barcoded Inaba and 8 barcoded Ogawa strains isolated from the libraries were sequenced individually to determine the sequence of their respective barcode. The strains were then mixed at similar proportions and incubated at different conditions indicated in Fig. 2b or in M9 at pH 6, pH 7 or pH 8 with or without 100 μg/ml polymyxin B or 100 μg/ml LL-37 (Fig. 2d). Cultures were then directly boiled and used for STAMPR PCR and analysis. All in vitro experiments were performed in 96-well plates in 150 μl at 37 °C for 16 hours.

## Barcode sequencing and analysis

Barcode sequencing and analysis were performed as previously described[27,29]. To amplify the barcoded region, PCR was performed after boiling of samples. Amplicons were verified by gel electrophoresis, cleaned using the Qiagen PCR Purification kit, and sequenced on a NextSeq1000. Sequencing reads were processed using the STAMPR pipeline (demultiplexing, trimming, mapping, and counting) to determine barcode frequencies[27,29]. If no reads were detected, values were substituted with 1, corresponding to the limit of detection.

## Slide agglutination

8 μl of saturated overnight cultures of the indicated *V. cholerae* strains were spotted on glass slides and mixed with 3 μl of either anti-Inaba or anti-Ogawa sera (BD Difco: Cat No.: D243047, lot: 0037338 for Inaba, or BD Difco: Cat No.: DF2431-47-0 lot: 9256006 for Ogawa). After 5-10 min of incubation at RT a photo of the results was taken using the Canon EOS 600D Rebel T3i camera.

## Microscopy

Samples from cultures of Inaba and Ogawa *V. cholerae* were stained with 1% Rhodamine B labeled Polymyxin B Ready Made Solution (Sigma) or 1% LL-37, Rhodamine conjugated (Rockland 000-000-M33) in PBS pH 7 or pH 8 for 30 minutes at 37 °C followed by 3 wash steps in PBS pH 7 or 8. The cells were then immobilized on 0.8% agar pads (in PBS of the respective pH) and imaged with a Nikon Ti2 Eclipse spinning disk confocal microscope using a 100× oil immersion lens with a numerical aperture of 1.45 and an Andor Zyla 4.2 Plus sCMOS monochrome camera. Image analysis was performed on ImageJ (2.14.0) software using custom macros.

## Software

Graphics and figures were prepared with BioRender, GraphPad Prism, and PowerPoint.

## Statistics & Reproducibility

In each case *n* represents individual measurements. No statistical method was used to predetermine sample size. No data were excluded from the analyses. The experiments were not randomized. The Investigators were not blinded to allocation during experiments and outcome assessment.

## Reporting summary

Further information on research design is available in the Nature Portfolio Reporting Summary linked to this article.

## Data availability

Source data are provided with this paper.

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

## Acknowledgements

We thank members of the Waldor lab for helpful discussions and feedback on the manuscript. We acknowledge our funding: Howard Hughes Medical Institute (M.K.W.). NIH grant R01 AI042347 (M.K.W.). Fellowship Zingl–2024HHMI (F.G.Z.). This article is subject to HHMI's Open Access to Publications policy. HHMI lab heads have previously granted a non-exclusive CC BY 4.0 license to the public and a sub-licensable license to HHMI in their research articles. Pursuant to those licenses, the author-accepted manuscript of this article can be made freely available under a CC BY 4.0 license immediately upon publication. This manuscript is the result of funding in whole or in part by the National Institutes of Health (NIH). It is subject to the NIH Public Access Policy. Through acceptance of this federal funding, NIH has been given the right to make this manuscript publicly available in PubMed Central upon the Official Date of Publication, as defined by NIH.

## Author contributions

Conceptualization: F.G.Z., B.F., M.K.W. Methodology: F.G.Z., B.F., D.R.L., A.A.M. Investigation: F.G.Z., B.F., D.R.L., A.A.M. Visualization: F.G.Z., A.A.M. Writing the original draft: F.G.Z., M.K.W. Reviewing and editing the manuscript: F.G.Z., B.F., D.R.L., A.A.M., M.K.W.

## Competing interests

The authors declare no competing interests.
