## [Transparent Peer Review file · Nature Communications]

***Vibrio cholerae* serotype impacts pathogenicity**

Corresponding Author: Dr Franz Zingl

Version 0:

Reviewer comments:

Reviewer #1

(Remarks to the Author)

This manuscript examines how the two *Vibrio cholerae* O1 serotypes, Ogawa (methylated O-antigen) and Inaba (unmethylated), differ in their pathogenesis. The authors use isogenic strain pairs that vary only at the *wbeT* locus to show that Ogawa is more likely to cause diarrhea in 8-day-old mice and achieves higher intestinal burdens in suckling mice. They further demonstrate that Ogawa seeds larger founding populations post-infection and has an approximately 10-fold lower ID_{50} . Finally, they show that Ogawa is more resistant to cationic antimicrobial peptides such as polymyxin B and LL-37 in vitro at pH 8.

The data are interesting and technically strong. However, the conclusions would be strengthened by direct in-vivo evidence that resistance to antimicrobial peptides is the causal mechanism underlying Ogawa's colonization advantage.

Major comments:

1. The manuscript requires greater detail about the age of the suckling mice used in Figure 1. The Methods state that colonization assays use CD-1 pups "post-natal day 5–8," which is a relatively large range as 5-day old mice are much more susceptible to colonization and disease compared to 8-day old suckling mice. This critical distinction is illustrated by the authors' own data: in Fig. 1b, most of the 8-day-old pups do not develop signs of diarrhea, which indicates that by day 8 they are no longer highly susceptible to disease and thus not suitable model of "cholera." There is no issue with using day 8 mice for the purposes of comparing pathogenesis of the two serotypes, but my concern lies with the definition of the model as a whole as "post-natal day 5–8," given the severe differences in disease susceptibility across that age range.

In Fig. 1b, the author's state that the diarrhea experiment is performed explicitly in 8-day-old pups inoculated with 2×10^5 CFU. Yet in the subsequent figure panels (Figure 1 c-g) the author's do not specify the age of the mice in the text or legend does for any panel.

- State the exact post-natal day for every in-vivo panel in Figure 1 (and elsewhere).
- Provide CFU data for the animals scored in Figure 1b. Without paired bacterial burdens it is impossible to know whether reduced diarrhea in the Inaba group reflects poorer colonization or another virulence-linked variable (e.g., founding-population size).
- Given that diarrheal disease in this model depends on both colonization and *ctxAB* expression, indicate whether Ogawa and Inaba strains express comparable cholera toxin levels in the 8-day model by qRT-PCR of *ctxA* or ELISA.
- Include a fluid-accumulation (FA) ratio or another quantitative disease metric in the younger-pup experiments that underlie panels 1c–g. If younger mice were indeed used in subsequent figure panels in Fig. 1, the author's should measure FA to assess how these serotypes impact disease in fully susceptible mice. This would help support the conclusion that these serotypes have a different pathogenicity potential.

2. A major criticism is the disconnect between the data showing Ogawa's in-vivo colonization advantage and the model that O-antigen methylation elevates colonization by shielding the bacterium from cationic antimicrobial peptides. In Figure 1, the authors demonstrate an in-vivo fitness advantage of Ogawa over Inaba, and in Figure 2 they show in vitro that Ogawa is more resistant to cationic AMPs than Inaba. However, while there is a correlation between colonization and increased resistance, the authors do not provide evidence for a direct causal link between Ogawa's colonization advantage and AMP resistance in vivo. To demonstrate a link, the authors would need to repeat the mixed-infection assays in AMP-deficient mice (e.g., *Camp*^{-/-} (LL-37/CRAMP-deficient) animals) and show that, in the absence of a major intestinal AMP, Ogawa's growth

advantage is significantly reduced. Without this direct in-vivo evidence, the conclusion that Ogawa's colonization advantage is due to AMP resistance is not supported by the current data.

Minor comments:

1. Although the CFU/Ns analysis in Figure 1f supports with the notion that Ogawa's advantage reflects improved survival of the founding population, it does not rule out a replication difference that could be masked by simultaneous killing or clearance. Given that this distinction underpins a main conclusion, the manuscript would be strengthened by a more direct measurement of in-vivo replication. One possible method would be an ori:ter (peak-to-trough) copy-number assay by multiplex qPCR (Mobley method, PMID 39605434) or shallow whole-genome sequencing (PMID 26229116). Demonstrating equivalent ori:ter ratios between the serotypes would provide concrete evidence that replication rates are indeed comparable.

2. Add reference (line 28)

3. Please, specify the pH on the experiments show in Fig. 2b with polymyxin B.

4. In the methods section it says AKI broth was used for cholera toxin induction. I couldn't find an experiment involving CT induction. Modify the text accordingly.

5. Specify carbon source on the M9 media in methods section

Reviewer #2

(Remarks to the Author)

In this study, Zingl and colleagues examines in a mouse model the potential differences in growth and pathogenicity between Ogawa and Inaba serotype of O1 serogroup O1 Vc. Findings were verified using several different clinical strains. The reason for the co-existence and cycling of serotypes has long been speculated on and typically presumed to be due to immunologic (or environmental) pressures. The knowledge that that Ogawa has a lower infectious dose and increased host colonization are important clues to our understanding of a) evolutionarily why these two serotypes exist/alternate and b) the role of serotypes in outbreaks in cholera-endemic areas.

1) These in vitro and in vivo studies propose a mechanism for the differences in clinical infection seen in mice, which are for the most part consistent with findings in humans, however this point is not mentioned in the manuscript. Human clinical manifestations of Ogawa vs. Inaba O1 Vc have been studied (PMID: 19883159). Here, Ogawa patients generally had more severe disease, required more intensive treatment, and had greater immune responses. It seems like including this clinical correlation in the manuscript would improve the premise and support what was found in mice.

2) It is also not mentioned that in humans, Ogawa is known to induce strain-specific immunity in humans (PMID: 21849288), while Inaba provides quite good cross protection. In the discussion, it is written "why do Inaba strains persist if they are less infectious and less efficient at intestinal colonization than Ogawa strains?" – Well, one reason is that Inaba-specific immunity is lost after a few years of Ogawa dominance in a population (see PMID: 32047137) – then – the question is that with Inaba present and people gaining heterologous protection, how/why does Ogawa recur? Maybe it gains a foothold due to low innuculum? In discussing the Ogawa/Inaba observed dynamics in humans, immunologic responses in humans to the different serotypes will need to be further described.

3) The sentence "However, at pH 8 but not pH 6 or 7, the Ogawa strains grew better than Inaba strains in both polymyxin B and LL-37, another cationic AMP produced in the human intestine" could be interpreted that polymyxin B is a human AMP, which is not the case. The relationship between these polymyxin and human AMPs should be clarified. While the mechanism of antimicrobial effect may be the same for LL-37 and polymyxin, what are their differences and is this mechanism common among AMPs? Why use polymyxin (instead of something human derived) for subsequent experiments? Use of a human AMP such as LL-37 in the confirmatory experiments would be more convincing to support the assertion that Ogawa is resistant to host-produced AMPs. Such as, with fluorescently labeled LL-37.

4) Could the mechanism of the severity of disease/inoculum differences of Ogawa be due to something else related to AMP? Such as another direct or indirect interaction of cationic peptides and the O-antigen of Ogawa cells?

Reviewer #3

(Remarks to the Author)

In this intriguing manuscript the authors address a longstanding question of whether there are differences in the abilities of Inaba and Ogawa *V. cholerae* serotypes to colonize hosts and cause disease. Using otherwise isogenic mutants that are unable to methylate the O-antigen, their data suggest that the Ogawa serotype colonizes better than Inaba and induces diarrhea in infant mice whereas Inaba does not. Ogawa seems to have a much lower infectious dose than Inaba. They exhibit no growth differences in vitro or in vivo after colonization has occurred. Using a panel of stressors. The authors found that Ogawa had a growth advantage in the presence of the antimicrobial peptide polymyxin-B. They propose that the O-antigen methylation in Ogawa prevents binding by cationic antimicrobial peptides. The ms is succinct and well written. Specific points:

1) Other than the paragraph between lines 30-43, it is unclear what strain background is being used in the experiments. I assume the Haitian strain? This should be clarified, especially for the experiments whose data are shown in Fig. 2.

Version 1:

Reviewer comments:

Reviewer #1

(Remarks to the Author)

I thank the authors for addressing many of my concerns and for performing additional experiments. However, while the added mouse experiments with buffered media are a step in the right direction, the data still do not provide a direct causal link between Ogawa's in-vivo colonization advantage and AMP resistance. The manuscript presents two separate findings: an in-vivo colonization advantage for Ogawa in the suckling mouse model, and an in-vitro survival advantage for Ogawa in the presence of specific antimicrobial peptides. These findings are not linked in vivo. The hypothesis that Ogawa's colonization advantage could be due to AMP resistance is reasonable, but the manuscript does not provide data that directly connect these observations.

Second, while the authors report earlier diarrhea in Ogawa-infected mice, the mechanism remains unclear. The *V. cholerae* virulence factor responsible for diarrhea in the suckling mouse model is cholera toxin. In their rebuttal, the authors state that they are not proposing Ogawa is intrinsically more toxigenic, yet no mechanism is offered to explain earlier diarrhea. This is a key finding reported in the manuscript, and the discrepancy (earlier disease without increased toxin production) should be addressed.

The authors show no difference in cholera-toxin gene expression between serotypes at 16 hours post-infection. However, if *wbeT* inactivation reduces survival and Ogawa seeds a larger founding population with similar replication and equal per-cell toxin output, wouldn't that mean more total toxin during early stages of infection in Ogawa-infected mice? Given the claim of earlier diarrhea in mice infected with Ogawa, additional earlier measurements would strengthen the conclusions.

To examine this, the authors should:

1. Report whether the "earlier diarrhea" in Figure 1B is statistically significant, including n, the statistical test, P values, etc.
2. Measure *ctxAB* transcripts and/or CT protein at earlier time points (e.g. 2-6 hours) during infection.
3. Quantify CFU and founding population at early time points (e.g. 2-6 hours) to determine whether Ogawa achieves higher early burdens that could explain earlier diarrhea.

Reviewer #2

(Remarks to the Author)

My reviewer concerns have been adequately addressed.

REVIEWER COMMENTS

Reviewer #1 (Remarks to the Author):

This manuscript examines how the two *Vibrio cholerae* O1 serotypes, Ogawa (methylated O-antigen) and Inaba (unmethylated), differ in their pathogenesis. The authors use isogenic strain pairs that vary only at the *wbeT* locus to show that Ogawa is more likely to cause diarrhea in 8-day-old mice and achieves higher intestinal burdens in suckling mice. They further demonstrate that Ogawa seeds larger founding populations post-infection and has an approximately 10-fold lower ID_{50} . Finally, they show that Ogawa is more resistant to cationic antimicrobial peptides such as polymyxin B and LL-37 in vitro at pH 8.

The data are interesting and technically strong. However, the conclusions would be strengthened by direct in-vivo evidence that resistance to antimicrobial peptides is the causal mechanism underlying Ogawa's colonization advantage.

We thank the reviewer for the careful read.

Major comments:

1. The manuscript requires greater detail about the age of the suckling mice used in Figure 1. The Methods state that colonization assays use CD-1 pups "post-natal day 5–8," which is a relatively large range as 5-day old mice are much more susceptible to colonization and disease compared to 8-day old suckling mice. This critical distinction is illustrated by the authors' own data: in Fig. 1b, most of the 8-day-old pups do not develop signs of diarrhea, which indicates that by day 8 they are no longer highly susceptible to disease and thus not suitable model of "cholera." There is no issue with using day 8 mice for the purposes of comparing pathogenesis of the two serotypes, but my concern lies with the definition of the model as a whole as "post-natal day 5–8," given the severe differences in disease susceptibility across that age range.

In our experience both P5 and P8 animals are highly susceptible to colonization and disease. However, to address the reviewer's concern, we removed the P8 data and performed additional experiments in P5 animals. In the revised manuscript all the data shown corresponds to P5 animals. This new work is shown revised Figure 2b and Supplementary Figure 2a,b.

In Fig. 1b, the author's state that the diarrhea experiment is performed explicitly in 8-day-old pups inoculated with 2×10^5 CFU. Yet in the subsequent figure panels (Figure 1 c-g) the author's do not specify the age of the mice in the text or legend does for any panel.

As noted above, all experiments are now done in P5 animals and the dose and age is now provided in the figure legends.

a. State the exact post-natal day for every in-vivo panel in Figure 1 (and elsewhere).

We added this information to all figure legends.

b. Provide CFU data for the animals scored in Figure 1b. Without paired bacterial burdens it is impossible to know whether reduced diarrhea in the Inaba group reflects poorer colonization or another virulence-linked variable (e.g., founding-population size).

It is not possible to obtain CFU values in an experiment like that shown in revised fig 1b and Supplementary Figure 2a,b because these experiments are designed to follow the natural history of disease. However, as shown in figures 1d and 1e, when animals are sacrificed at 18 hours after infection, Ogawa serotype strains have greater CFU and FP respectively. We are not claiming that the higher virulence (diarrhea) exhibited by Ogawa strains is a property that is distinct from Ogawa's greater colonization (infectivity) capacity. Ultimately after 25-30 hours, CFU in intestinal homogenates is similar, representing the maximal carrying capacity of P5 intestines.

c. Given that diarrheal disease in this model depends on both colonization and *ctxAB* expression, indicate whether Ogawa and Inaba strains express comparable cholera toxin levels in the 8-day model by qRT-PCR of *ctxA* or ELISA.

The reviewer raises an excellent point. We compared CT expression of both serotypes during in vivo colonization via qrtPCR and after growth in AKI media, which is a proxy for in vivo virulence expression via ELISA. We present this new dataset in the revised manuscript in a new Supplementary Figure 3. Our results show that *ctx* expression per cell is similar between the two serotypes in AKI and in vivo. These data support the idea that Ogawa's greater colonization account for its greater virulence. This point is now raised in the manuscript as well.

d. Include a fluid-accumulation (FA) ratio or another quantitative disease metric in the younger-pup experiments that underlie panels 1c–g. If younger mice were indeed used in subsequent figure panels in Fig.

1, the author's should measure FA to assess how these serotypes impact disease in fully susceptible mice. This would help support the conclusion that these serotypes have a different pathogenicity potential.
Thank you, we have now measured weight of the diarrheal discharge and mouse weight loss as metrics of disease. This data is now included in a new Supplementary Figure 2a,b.

2. A major criticism is the disconnect between the data showing Ogawa's in-vivo colonization advantage and the model that O-antigen methylation elevates colonization by shielding the bacterium from cationic antimicrobial peptides. In Figure 1, the authors demonstrate an in-vivo fitness advantage of Ogawa over Inaba, and in Figure 2 they show in vitro that Ogawa is more resistant to cationic AMPs than Inaba. However, while there is a correlation between colonization and increased resistance, the authors do not provide evidence for a direct causal link between Ogawa's colonization advantage and AMP resistance in vivo. To demonstrate a link, the authors would need to repeat the mixed-infection assays in AMP-deficient mice (e.g., Camp-/- (LL-37/CRAMP-deficient) animals) and show that, in the absence of a major intestinal AMP, Ogawa's growth advantage is significantly reduced. Without this direct in-vivo evidence, the conclusion that Ogawa's colonization advantage is due to AMP resistance is not supported by the current data.

The reviewer raises a valid point. However, the camp -/- mice are only available as frozen sperm and to carry out this experiment would take >6 months; furthermore, since mice have many additional AMPs besides CRAMP it is not clear that the experiment will adequately address the hypothesis that methylation of the O-antigen protects from CAMP activity at pH 8. To begin to address the reviewer's concern, we leveraged our in vitro data that showed that at pH 6 or pH 7 Ogawa and Inaba are similarly affected by CAMPS. Thus, we carried out competitive infection experiments with buffered inputs at pH 6 or pH 7. Indeed, we saw that buffering the inoculum at these pHs largely eliminated the colonization advantage of the Ogawa serotype. These Data are now included in a new Figure 2a.

Minor comments:

1. Although the CFU/Ns analysis in Figure 1f is supports with the notion that Ogawa's advantage reflects improved survival of the founding population, it does not rule out a replication difference that could be masked by simultaneous killing or clearance. Given that this distinction underpins a main conclusion, the manuscript would be strengthened by a more direct measurement of in-vivo replication. One possible method would be an ori:ter (peak-to-trough) copy-number assay by multiplex qPCR (Mobley method, PMID 39605434) or shallow whole-genome sequencing (PMID 26229116). Demonstrating equivalent ori:ter ratios between the serotypes would provide concrete evidence that replication rates are indeed comparable.

We agree with the reviewer that the replication rate could potentially be masked by additional effects. We modified the language to clarify that we are talking about net expansion (which takes simultaneous killing or clearance into account) and not replication rate. Similar net replication indicates that the number of cells originating from one founder is similar for Inaba and Ogawa indicating that the difference in final burden originated from a difference in the number of founders.

2. Add reference (line 28)

Done.

3. Please, specify the pH on the experiments show in Fig. 2b with polymyxin B.

The pH was non buffered in this experiment. We added a clarifying statement. The starting pH of the M9 media was 6.92.

4. In the methods section it says AKI broth was used for cholera toxin induction. I couldn't find an experiment involving CT induction. Modify the text accordingly.

Thank you we now show the AKI experiments.

5. Specify carbon source on the M9 media in methods section

Done.

Reviewer #2 (Remarks to the Author):

In this study, Zingl and colleagues examines in a mouse model the potential differences in growth and pathogenicity between Ogawa and Inaba serotype of O1 serogroup O1 Vc. Findings were verified using several different clinical strains. The reason for the co-existence and cycling of serotypes has long been speculated on and typically presumed to be due to immunologic (or environmental) pressures. The knowledge that that Ogawa has a lower infectious dose and increased host colonization are important clues to our understanding of a) evolutionarily why these two serotypes exist/alternate and b) the role of serotypes in outbreaks in cholera-endemic areas.

We thank the reviewer for the careful read and positive comments.

1) These in vitro and in vivo studies propose a mechanism for the differences in clinical infection seen in mice, which are for the most part consistent with findings in humans, however this point is not mentioned in the manuscript. Human clinical manifestations of Ogawa vs. Inaba O1 Vc have been studied (PMID: 19883159). Here, Ogawa patients generally had more severe disease, required more intensive treatment, and had greater immune responses. It seems like including this clinical correlation in the manuscript would improve the premise and support what was found in mice.

Thanks for this important reference! We have added the potential human correlate to the discussion.

2) It is also not mentioned that in humans, Ogawa is known to induce strain-specific immunity in humans (PMID: 21849288), while Inaba provides quite good cross protection. In the discussion, it is written “why do Inaba strains persist if they are less infectious and less efficient at intestinal colonization than Ogawa strains?” – Well, one reason is that Inaba-specific immunity is lost after a few years of Ogawa dominance in a population (see PMID: 32047137) – then – the question is that with Inaba present and people gaining heterologous protection, how/why does Ogawa recur? Maybe it gains a foothold due to low innaculum? In discussing the Ogawa/Inaba observed dynamics in humans, immunologic responses in humans to the different serotypes will need to be further described.

Thanks, we have mentioned the immunologic aspects of the serotypes in the discussion.

3) The sentence “However, at pH 8 but not pH 6 or 7, the Ogawa strains grew better than Inaba strains in both polymyxin B and LL-37, another cationic AMP produced in the human intestine” could be interpreted that polymyxin B is a human AMP, which is not the case. The relationship between these polymyxin and human AMPs should be clarified. While the mechanism of antimicrobial effect may be the same for LL-37 and polymyxin, what are their differences and is this mechanism common among AMPs? Why use polymyxin (instead of something human derived) for subsequent experiments? Use of a human AMP such as LL-37 in the confirmatory experiments would be more convincing to support the assertion that Ogawa is resistant to host-produced AMPs. Such as, with fluorescently labeled LL-37.

We clarified the difference between Polymyxin and LL-37 and now show experiments using labeled LL-37 in the new fig 2f.

4) Could the mechanism of the severity of disease/inoculum differences of Ogawa be due to something else related to AMP? Such as another direct or indirect interaction of cationic peptides and the O-antigen of Ogawa cells?

The reviewer is correct. Additional mechanisms besides blocking of hydrogen bonds are possible. Such mechanisms might include steric hinderance of interactions or changes in polarity that reduce interactions with antimicrobial peptides. However, we favor the blocking of hydrogen bonds as it explains the difference between pH 7 and pH 8.

Reviewer #3 (Remarks to the Author):

In this intriguing manuscript the authors address a longstanding question of whether there are differences in the abilities of Inaba and Ogawa V. cholerae serotypes to colonize hosts and cause disease. Using otherwise isogenic mutants that are unable to methylate the O-antigen, their data suggest that the Ogawa serotype colonizes better than Inaba and induces diarrhea in infant mice whereas Inaba does not. Ogawa seems to have a much lower infectious dose than Inaba. They exhibit no growth differences in vitro or in vivo after colonization has occurred. Using a panel of stressors. The authors found that Ogawa had a growth advantage in the presence of the antimicrobial peptide polymyxin-B. They propose that the O-antigen methylation in Ogawa prevents binding by cationic antimicrobial peptides. The ms is succinct and well written.

We thank the reviewer for the careful read and positive comments.

Specific points:

1) Other than the paragraph between lines 30-43, it s unclear what strain background is being used in the

experiments. I assume the Haitian strain? This should be clarified, especially for the experiments whose data are shown in Fig. 2.

We added clarifying statements to the figure legends.

Reviewer #1 (Remarks to the Author):

I thank the authors for addressing many of my concerns and for performing additional experiments. However, while the added mouse experiments with buffered media are a step in the right direction, the data still do not provide a direct causal link between Ogawa's in-vivo colonization advantage and AMP resistance. The manuscript presents two separate findings: an in-vivo colonization advantage for Ogawa in the suckling mouse model, and an in-vitro survival advantage for Ogawa in the presence of specific antimicrobial peptides. These findings are not linked in vivo. The hypothesis that Ogawa's colonization advantage could be due to AMP resistance is reasonable, but the manuscript does not provide data that directly connect these observations.

As we stated in response to this valid concern in the original review, obtaining "the camp +/- mice are only available as frozen sperm and to carry out this experiment would take >6 months; furthermore, since mice have many additional AMPS besides CRAMP it is not clear that the experiment will adequately address the hypothesis that methylation of the O-antigen protects from CAMP activity at pH 8.". Our new competitive infection data with buffered inputs supports our model. However, we never claimed our data demonstrates a direct link between Ogawa's higher cAMP resistance and the higher virulence. Furthermore, we added a statement in the discussion that explicitly states that this idea remains a hypothesis.

Second, while the authors report earlier diarrhea in Ogawa-infected mice, the mechanism remains unclear. The *V. cholerae* virulence factor responsible for diarrhea in the suckling mouse model is cholera toxin. In their rebuttal, the authors state that they are not proposing Ogawa is intrinsically more toxigenic, yet no mechanism is offered to explain earlier diarrhea. This is a key finding reported in the manuscript, and the discrepancy (earlier disease without increased toxin production) should be addressed.

Thank you for the opportunity to clarify. Since our results show that *ctx* expression per cell is similar between the two serotypes in AKI and in vivo, they support the idea that Ogawa's greater colonization account for its greater virulence. We propose that the greater CFU burden explains the greater or earlier diarrhea. In vivo measures of cholera toxin expression were normalized to the 16s RNA and thus in essence normalize to CFU. We added a new sentence to the discussion making this point clear "Accordingly, we propose that the elevated virulence observed during Ogawa infection results from its enhanced colonization capacity."

The authors show no difference in cholera-toxin gene expression between serotypes at 16 hours post-infection. However, if *wbeT* inactivation reduces survival and Ogawa seeds a larger founding population with similar replication and equal per-cell toxin output, wouldn't that mean more total toxin during early stages if infection in Ogawa-infected mice? Given the claim of earlier diarrhea in mice infected with Ogawa, additional earlier measurements would strengthen the conclusions.

Yes, we agree. Greater *V. cholerae* burden leads to higher toxin output at early stages of infection.

To examine this, the authors should:

1. Report whether the "earlier diarrhea" in Figure 1B is statistically significant, including n, the statistical test, P values, etc.

The earlier diarrhea in figure 1B reaches statistical significance using a two-sided Mantel-Cox test. We now report this in the figure.

2. Measure *ctxAB* transcripts and/or CT protein at earlier time points (e.g. 2-6 hours) during infection.

In our view, the explanation that the greater burden of Ogawa vs Inaba explains the earlier onset and greater diarrhea is plausible and carrying out additional animal experiments is not justified. Since there is no detectable difference in diarrhea until >20 hours post infection, measurements of CT expression before that time would not likely be fruitful.

3. Quantify CFU and founding population at early time points (e.g. 2-6 hours) to determine whether Ogawa achieves higher early burdens that could explain earlier diarrhea.

Understanding the kinetics of colonization and cholera toxin production go beyond the scope of this manuscript.